

# Fusion of Lagrangian drifter data and numerical model outputs for improved assessment of turbulent dispersion

Sloane Bertin[1], Alexei Sentchev[1], Elena Alekseenko[1]

[1]Laboratory of Oceanology and Geosciences, UMR8187, Univ. Littoral Côte d'Opale, CNRS, Univ. Lille, IRD,
Wimereux, 62930, France

*Correspondence to*: Sloane Bertin (sloane.bertin@univ-littoral.fr)

**Abstract.** Transport and dispersion processes in the ocean are crucial, as they determine the lifetime and fate of biological and chemical quantities drifting with ocean currents. Due to the complexity of the coastal ocean environment, numerical circulation models have difficulties to accurately simulate highly turbulent flows and
dispersion processes, especially in highly energetic tidal basins such as the eastern English Channel. A method of improving the results of coastal circulation modeling and tracer dispersion in the Dover Strait is proposed. Surface current velocities derived from Lagrangian drifter measurements in November 2020 and May 2021 were optimally interpolated in time and space to constrain a high-resolution coastal circulation MARS model, with careful attention given to selecting ensemble members composing the model covariance matrix. The space-time velocity covariances
derived from model simulations were utilized by the Optimal Interpolation algorithm to determine the most likely evolution of the velocity field under constraints provided by Lagrangian observations and their error statistics. The accuracy of the velocity field reconstruction was evaluated at each time step. The results of the fusion of model outputs with surface drifter velocity measurements show a significant improvement (by ~50%) of the model capability to simulate the drift of passive tracers in the Dover Strait. Optimized velocity fields were used to quantify the absolute
dispersion in the study area. The implications of these results are important, as they can be used to improve existing decision-making support tool or design new tools for monitoring the transport and dispersion in coastal ocean environment.

## 1 Introduction

Despite a progress achieved recently in simulating the large-scale O (>100 km) and mesoscale O(10-100 km) variability of ocean currents (e. g., Jansen et al., 2019; Zanna and Bolton, 2020), accurately reconstructing small scale features of ocean circulation remains challenging. While circulation models have spatial resolutions of hundreds of meters (Callies et al., 2015; Grist et al., 2021; Nguyen-Duy et al., 2021), the lack of direct observations makes difficult the validation of modeling



results at sub-mesoscale O(1-10 km). In fact, the sub-mesoscale features of the ocean circulation are difficult to measure by
existing observing systems and, when measured, their resolution rarely matches that obtained by modeling.

Numerous studies have emphasized the significance of sub-mesoscale variability of ocean circulation which appears highly energetic and ageostrophic. Such sub-mesoscale motions have a notable impact on energy cascade and energy dissipation in the ocean (Ferrari and Wunsch, 2009), as well as on horizontal transport of suspended matter (Aleskerova et al., 2019) or budgets of physical and biological quantities (Uchida et al., 2020). Keerthi et al., (2022) demonstrated that the annual
changes in phytoplankton biomass in the Gulf of Mexico are driven by small-scale physical processes (eddies, atmospheric storms, etc), that control growth and spatial distribution of phytoplankton, and are influenced by the exchange of energy and matter between the atmosphere and the ocean. Both models and observations indicate that the dispersal rate in the presence of sub-mesoscale turbulence can easily exceed the mesoscale dispersal rate in the geostrophic current by an order of magnitude (Haza et al., 2008; Poje et al., 2014).

An incomplete knowledge of forcings in combination with the complexity of coastal environment, which includes a complex shoreline, river mouths, beaches, submarine banks, etc, presents a real challenge for numerical modeling. As a result, circulation models have difficulties in simulating a highly turbulent coastal flow at sub-mesoscale. Hence, it is important to develop techniques that can improve the model skill to reconstruct the water circulation and dispersion processes in coastal environment in a simple and efficient way.

The current study employs a method of Optimal Interpolation (OI) of Lagrangian observations using a high-resolution regional circulation model as a background. Lagrangian observations of current velocities are used to correct the model trajectories in an optimal way. Pioneered by Gandin (1963), and applied in atmospheric modeling, the OI has been widely used in different fields of geosciences for mapping the sea surface temperature (Bretherton et al., 1976), modeled current velocity optimization ( Molcard et al., 2003; Sentchev and Yaremchuk, 2015), or topography optimization (Wu et al., 2021).

Compared to other approaches to optimizing ocean circulation such as variational methods (e. g., Kalnay, 2002; Sentchev and Yaremchuk, 1999; Wikle, 2005), the OI has several advantages. Firstly, the method is straightforward to implement and ensures a reasonable balance between the computational complexity and statistical consistency of the model-data misfits. Second, the accuracy of the reconstructed velocity field can be inexpensively evaluated at every time step of the model.

The use of OI of observations leads to a significant improvement of the current velocity fields and velocity gradients, which
are often inadequately represented in the models due to their low resolution or intrinsic limitations. Therefore, the turbulent dispersion appears also affected by these limitations. Many studies focused on the investigation of processes that influence the dispersion in the ocean, such as tidal motions (Meyerjürgens et al., 2020), waves (Weichman and Glazman, 2000), and the variability of ocean circulation (Haza et al., 2008; LaCasce and Ohlmann, 2003; Lumpkin and Elipot, 2010). The present study aims to quantify the effect of current velocity optimization on the dispersion rate of passive tracers in a tide dominated
region - the Dover Strait, in the eastern English Channel (EEC).

The paper is organized as follows: Section 2 provides a general presentation of the study area and the data used. Section 3 provides a detailed description of the methods utilized in this study. The results of Optimal Interpolation of Lagrangian





measurements and characterization of dispersion processes are presented in Section 4. Furthermore, a technique for correcting the wind-driven velocity component of surface currents is proposed in this section. Discussion and conclusions

are presented in Section 5 and 6, respectively.

## 2 Study site and the data

### 2.1 Study site and hydrodynamic conditions

The study was carried out in the Dover Strait, a shallow water region of the northwest European continental shelf connecting

the English Channel to the North Sea (Fig.1a). The region is characterized by highly irregular bathymetry, with depth not exceeding 65 m and the presence of many sandbanks, roughly oriented in the dominant current direction, with depth only of a few meters at low tide.

Tidal motions of semi-diurnal period dominate the local circulation. The tidal range in Boulogne, located on the eastern coast of France (Fig. 1a), is close to 9 meters and the current speed can reach 2 m/s during spring tide. The tidal stream loses

around 20% of its intensity during ebb tide and the sea surface height and velocity are characterized by a pronounced asymmetry. A large-scale circulation in the North Atlantic Ocean generates sea level difference driving a weak residual flow from the Atlantic Ocean towards the North Sea. This is another remarkable feature of the local hydrodynamics. The order of magnitude of the tidal residual currents in the Dover Strait ranges between 5 and 10 cm/s (Lazure and Desmare, 2012). The spatial variability of residual currents is caused by topographic features of the English Channel that constrain tidal wave

propagation (e. g., Sentchev and Yaremchuk, 2007).

The wind significantly affects the local circulation. Southwestern winds can increase the average eastward flow while northwestern and northeastern winds can reduce the tidal flow opposing the wind and even reverse it (e. g., Lazure and Desmare, 2012). The freshwater input from rivers located on the French coast (the Seine, the Somme, the Authie rivers) has only a little influence on the water circulation in the study area.




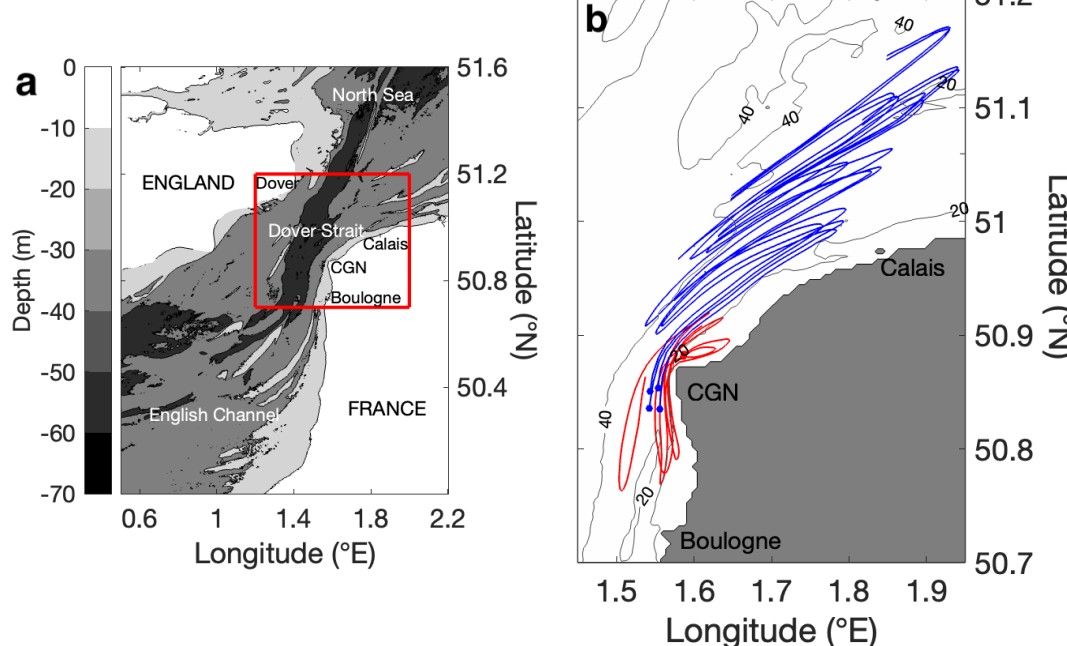

**Figure 1. (a) Modeling domain. Bathymetry shown by grey shading. Red square delimits the region where the Lagrangian measurements were performed. (b) Trajectories of Lagrangian drifters released during two field surveys on November 26, 2020 (red) and on May 10, 2021 (blue). Geographic names used in the text are also shown.**


## 2.2 Current velocity measurements

A total of six Lagrangian drifters were deployed in the Dover Strait during two periods of time, under relatively calm to moderate winds (mean wind speed less than 8 m/s) and waves not exceeding 1 m height. During the first survey, referred to hereafter as S1, two surface drifters were released north of the CGN (Fig. 1b, red trajectories) for 26-hours period, from

November 26, 2020, 8:30 UTC to November 27, 2020, 11:00 UTC. At the release, the two drifters were separated by 250 m. The survey was performed under mean tide conditions and northeastern wind of 4 m/s, on average. The two drifters of S1 will be referred to hereafter as S1-1 and S1-2. During the second survey, referred to hereafter as S2, four surface drifters were deployed west of the CGN (Fig. 1b, blue trajectories) for 46-hours period, from May 10, 2021, 9:15 UTC to May 12, 2021, 07:30 UTC. The drifters formed a rectangle of size 1.3 by 2 km. The survey was performed under spring tide

conditions and stronger southwestern winds of 6 m/s, on average, with gusts up to 12 m/s. The four drifters of S2 will be referred to hereafter as S2-1, S2-2, S2-3, and S2-4.





Two types of buoys were used: coastal Nomad surface buoys manufactured by SouthTek (https://www.southteksl.com/) and drifters manufactured at the lab. The latter featuring a cylindrical PVC hull of 0.6 m long and 0.1 m in diameter weighted in its lower part. A thin square plate of 0.3 x 0.3 m in size was installed in the upper part of the hull to assure better stability in the vertical and reduce the pitch. The drifters were equipped with Smartone GPS/satellite transceiver of Global Star satellite network. All drifters were equipped with an anchor of 0.5 m long positioned in the water layer between 0.8 and 1.3 m depth, allowing them to drift with surface currents.

Observed surface current velocities were estimated from the drifter trajectories with a timestep of 15 min, which was a nominal period of drifter positioning via GPS. During S1, the mean drifter velocity was 0.8 m/s. The maximum speed of 1.6 m/s was reached during peak flood flow and observed north of the CGN. The minimum speed of 0.1 m/s was reached 2-h after peak ebb flow and observed south of the CGN. During S2, the mean and maximum drifter velocities were found to be 1 m/s and 2.1 m/s respectively (Fig. 1a).

## 2.3 Current velocity from numerical model

Numerical simulations were conducted using a two-dimensional (2D) Model for Application at Regional Scale (MARS) (Lazure and Dumas, 2008). Model fields, including surface elevation and depth averaged velocity, are available online at "Modeling and Analysis for Coastal Research" (MARC) project website (https://marc.ifremer.fr). MARS is a hydrodynamic model solving a system of incompressible Navier-Stokes equations using the Boussinesq approximation and the hydrostatic assumption. The turbulence scheme implemented in the model is detailed in Gaspar et al. (1990). The modeling domain covers an area much larger than the EEC, in order to properly reproduce storm surges generated at greater distance in the Atlantic ocean or in the North Sea (Idier et al., 2012). The model grid size was 250 m and temporal resolution 15 min. The accuracy of MARC simulations has been validated in February 2010 in 19 tide gauges locations on the French coast, including Boulogne and Calais. The resulting mean root mean square errors are of 11 cm for the tide alone and 16 cm for the sea level (tide and surge).

The bathymetry was provided by the SHOM (French Hydrographic and Oceanographic Service). Tidal boundary conditions were taken from the global tidal model FES2004 (Lyard et al., 2006). Intertidal areas were simulated with a wetting and drying scheme. The drag coefficient used for wind effect parametrization is the variable Charnock coefficient from WWIII wave model (Ardhuin et al., 2011). Originally represented on an Arakawa C-grid, surface currents were interpolated on the Arakawa A-grid (Arakawa and Lamb, 1977) for further analysis and optimization of model velocities. The model used in the analysis will be referred to hereafter as M2D.

The present study is based on two one-year long model runs covering the period from January to December 2020 containing S1, and from January to December 2021 containing S2. These long periods are useful for proper selection of ensemble members required for the covariance matrix calculation for OI.



## 2.4 Wind data

Meteorological data (wind, temperature, humidity, and atmospheric pressure) are used as forcing of M2D. The data were provided by Arpege (Action de Recherche Petite Echelle Grande Echelle) operational atmospheric model of Météo-France with 5 km spatial and 1 h time resolution.

The meteorological data from the model were compared to *in situ* measurements collected at meteorological station in Boulogne and Calais during the year of model simulations. The time and space averaged difference between the observed and modeled wind speed was found to be 1.7 m/s and 0.9 m/s for the surveying periods, giving confidence in the model wind data.

Figure 2 shows the wind rose for each survey from Arpege model. Two dominant wind regimes were observed during the surveyed days. During S1, the wind direction was towards the southwest, and the speed did not exceed 5 m/s with the mean value of 4 m/s. During S2, the wind had an opposite direction, and the speed varied within the range of 4-9 m/s, with the mean speed 6 m/s and the maximum speed 11 m/s.

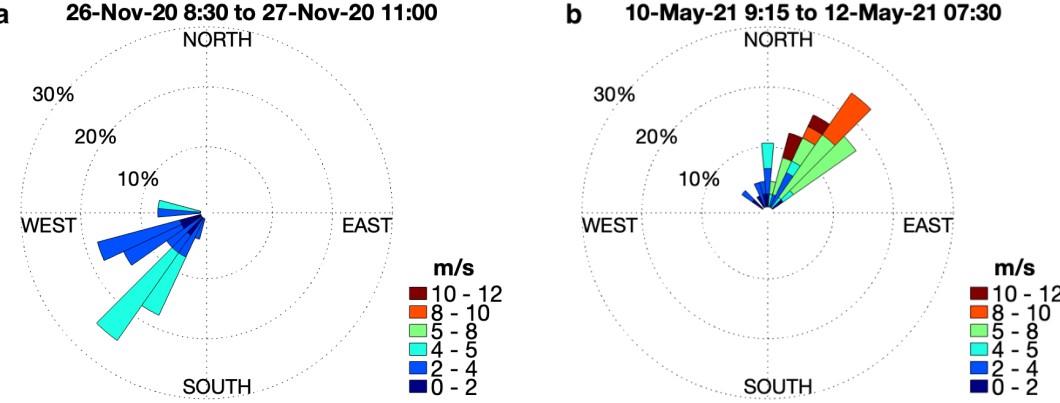

**Figure 2. Wind roses for two survey periods: S1 (a) and S2 (b) from Arpege atmospheric model (hourly data) spatially-averaged over the study region.**

## 3 Methodology

### 3.1 Optimal Interpolation of velocity measurements

One of the methods used to constrain the numerical model outputs by observations is the Optimal Interpolation of observations. It provides an estimate of the state of the ocean dynamics by performing a weighted Least Squares fit of a





background model field to observations. In general, observations are available at irregularly distributed points and are
assumed to be imperfect, i. e., each observation being affected by an uncertainty (observation error). It is assumed that the
observation error is not correlated with the model error.

In the OI method, a correction of a background velocity field $\boldsymbol{u}_m(\boldsymbol{x}, t)$, provided by a numerical model on a regular grid, is
done using a linear combination of the weighted differences between the background model trajectory $\boldsymbol{u}_m$, and the observed
velocities $\boldsymbol{u}_i^*$ at point $i$ (Bretherton et al., 1976; Sentchev and Yaremchuk, 2015; Thiébaux and Pedder, 1987). Weights
chosen for minimization of the mean square difference between the observed and background velocities are a combination of
model and observation covariances. With the space-time covariance matrices of the model $\boldsymbol{B} = \langle \boldsymbol{u}_m(\boldsymbol{x}, t)\boldsymbol{u}_m(\boldsymbol{x}', t') \rangle$ and
observations $\boldsymbol{R}_{ij} = \langle \boldsymbol{u}_i^* \boldsymbol{u}_j^* \rangle$, the optimized velocities $\boldsymbol{u}_{OI}$ are computed as follows:

$$\boldsymbol{u}_{OI} = \boldsymbol{u}_m + \sum_{ij} \boldsymbol{B} \boldsymbol{H}_j^T \left( \boldsymbol{H}_i \boldsymbol{B} \boldsymbol{H}_j^T + \boldsymbol{R}_{ij} \right)^{-1} (\boldsymbol{H}_i \boldsymbol{u}_m - \boldsymbol{u}_i^*) . \tag{1}$$

Here, $\boldsymbol{H}_i$ corresponds to a linear operator projecting gridded velocity values from the apexes of the model grid cell onto the
$i^{th}$ observation point.

The quality of the interpolation scheme is quantified by estimating the mean relative difference between the observations $\boldsymbol{u}^*$
and optimized model velocities $\boldsymbol{u}_{OI}$ as follows:

$$\varepsilon_{OI} = \sqrt{\frac{\sum_i (\boldsymbol{H}_i \boldsymbol{u}_{OI} - \boldsymbol{u}_i^*)^2}{\sum_i (\boldsymbol{u}_i^*)^2}} . \tag{2}$$

The relative error of the initial model, $\varepsilon_m$, is quantified in the same way.

An important assumption underlying the OI method is that the background field, also called the "first guess", is a good
approximation of the truth. Thus, the computation of $\boldsymbol{B}$ and $\boldsymbol{u}_m$ is crucial. The background velocity fields were provided by
the model at observation time step.

For estimation of model covariances, a variable number of model trajectories (26-h long for S1 and 46-h long for S2),
referred to as ensemble members, were used. The sensitivity test of OI to the number of ensemble members used was
performed and the results are presented in Section 4.1. Three approaches were used in selecting the ensemble members. The
first and easiest way is extracting them from the model on the days surrounding the survey, and by respecting the conditions
(wind and tidal conditions) observed during the surveying days. A total of 7 ensemble members were selected using this
approach.





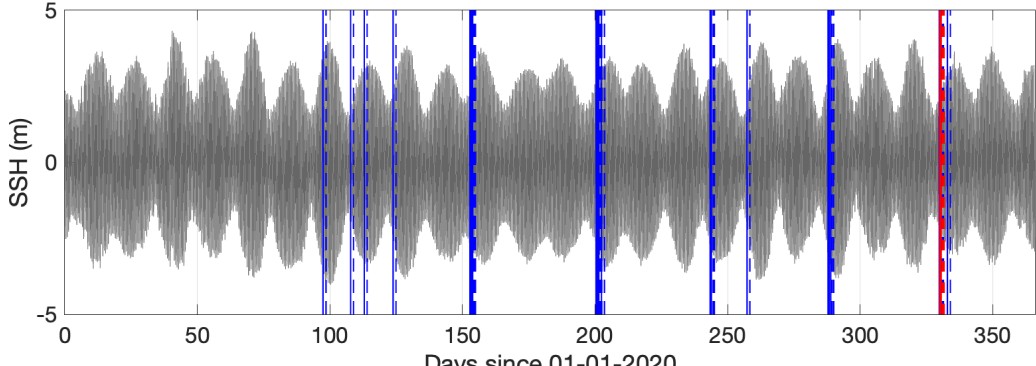

**Figure 3. A total of 31 ensemble members (in blue) extracted from the model simulation of circulation in the Dover Strait using the**
**realistic forcing in 2020. S1 period is shown in red. Sea surface height (SSH) from the model run is given in dark grey.**

However, for an evolving ocean state, a large number of ensemble members might be required to represent the flow field
evolution with statistical significance. Therefore, in the second step, ensemble members were extracted from the one-year
long model simulation (January to December 2020) containing S1, by searching for wind and tidal conditions similar to
those observed during the surveying period. By accepting a variation of the average wind speed and direction within the
range $\pm$ 2 m/s and $\pm$ 45° respectively, for a given tidal stage, a total of 31 ensemble members were selected. In a third step,
a more restrictive criterion of the range of variation of the wind, for instance, $\pm$ 1 m/s and $\pm$ 45° , allowed to obtain 11
ensemble members. Figure 3 shows a chronology of ensemble members selected for OI of velocity observations during S1.
Each ensemble member represents a 26-h long model run.


### 3.2 Lagrangian trajectories reconstruction

In addition to the relative error $\varepsilon$, the quality of the interpolation scheme can be assessed by estimating the separation
distance $d$ between the real trajectories of drifting buoys and that provided by the model. The latter were reconstructed using
OceanParcels Lagrangian framework (https://oceanparcels.org/). Virtual particles were seeded at the time and location of the
real drifters at the release, then advected during a given period of time using horizontal forward Euler method without
diffusion, giving their time-dependent position $x(t)$ and $y(t)$. The separation distance $d$, estimated at 15-min time step and
averaged over drifters, is a commonly used metric which shows how good are the drift trajectories reconstructed by the
initial model (the separation distance $d_m$ ), or the optimized model (the separation distance $d_{OI}$).



### 3.3 Correction of the wind-induced current velocities

The ocean-atmosphere coupling is difficult to reproduce correctly especially in coastal regions. In fact, the wind generates surface Ekman currents directed 45° to the right of the wind at the sea surface. It is assumed that this wind-driven velocity can attain 1-3% of the wind speed at 10 m height (Jenkins, 1987; Weber, 1983).

Imperfect representation of the wind-driven velocity in the model can be improved by using velocity measurements by surface drifters. Let us assume that the flow velocity can be decomposed in two parts: the optimally interpolated velocity $\boldsymbol{u}_{OI}$

and an additional correction, $\boldsymbol{c}\overline{\boldsymbol{U}}_{10}$, where $\overline{\boldsymbol{U}}_{10}$ is the wind velocity vector at 10 m height, averaged over the study area and the survey period, and $\boldsymbol{c} = diag(c_x, c_y)$ is a diagonal 2 x 2 matrix whos diagonal elements are estimated by minimizing the cost function:

$$\mathcal{J}(c) = [\boldsymbol{X}^* - (\boldsymbol{X}_0 + \sum_k (\boldsymbol{u}_{OI})_k \Delta t + \boldsymbol{c}\overline{\boldsymbol{U}}_{10}\Delta t)]^2 \to \frac{min}{c}, \qquad (3)$$

here $\boldsymbol{X}^*$ is a sequence of drifter coordinates at $\Delta t = 15\ min$ time stepping, $\boldsymbol{X}_0$ is the coordinate at the release, and summation

is performed over a drifter trajectory. The expression in parenthesis (.) represents a virtual drifter trajectory after correction for the wind effect. The coefficients $(c_x, c_y)$ were estimated for each drifter trajectory and then averaged. This implies an assumption of a stationary wind (mainly wind direction) that was supported by observations at meteorological stations during the surveying periods S1 and S2. Correcting the wind-induced velocity enables better reconstruction of the optimized velocity fields denoted hereinafter by $\boldsymbol{u}_{cor}$. The relative error of the model after performing the wind-induced velocity

correction is computed using (2), after replacing $\boldsymbol{u}_{OI}$ by $\boldsymbol{u}_{cor}$. The separation distance between the observed trajectories and trajectories reconstructed from the model after performing wind-induced velocity correction is referred to as $d_{cor}$.

### 3.4 Absolute dispersion

The absolute dispersion $A^2$ is defined as the variance of particle spreading with respect to the mean coordinate of particles in

a cluster (the barycenter). In two-dimensions, the dispersion is generally estimated along $x$ and $y$ axis (Berti et al., 2011; Enrile et al., 2019). But in this study, to better account for the dominant flow direction, the variance is computed in the direction of the maximum spreading of particles and in the perpendicular direction, providing two quantities $A_1^2$ and $A_2^2$. They represent the major and minor axis of the deformation tensor and are estimated by applying the Principal Component Analysis (PCA) to particle distribution at each time step (Emery and Thomson, 2004). As the tidal flow direction in shallow-

water basins is generally constrained by local topography and coastline orientation, the ellipse orientation (major axis) gives the dominant flow direction. The ellipse orientation $\theta$, and the variances $A_1^2$ and $A_2^2$, were computed as follows (Emery and Thomson, 2004):

$$\theta = \frac{1}{2} tan^{-1} \left[ \frac{2\overline{x'y'}}{\overline{x'^2 - y'^2}} \right], \qquad (4)$$



$$\begin{bmatrix} A_1^2 \\ A_2^2 \end{bmatrix} = \frac{1}{2} \left\{ \left( \overline{x'^2} + \overline{y'^2} \right) \pm \left[ \left( \overline{x'^2} - \overline{y'^2} \right)^2 + 4 \left( \overline{x'y'} \right)^2 \right]^{\frac{1}{2}} \right\}. \tag{5}$$

Here, $\overline{x'^2}$ and $\overline{y'^2}$ stand for variances of particle coordinates along the eastward $x$ and northward $y$ axes respectively.

## 4 Results

### 4.1 Model velocities after Optimal Interpolation of the Lagrangian observations

Figure 4 shows the results of Lagrangian drifter velocities interpolation for S2. The largest number of ensemble members,
31, was used in interpolation. The discrepancy between the initial and optimally interpolated velocities during peak flood
and ebb flow (Fig. 4, color shading) varies in space with lower values found south of the CGN, for both flood and ebb flow,
and the largest value (~0.5 m/s) found in the northern part of the Strait, close to the U.K. coast. In the French sector of the
Strait, the discrepancy attains 0.2 m/s over the sandbanks. However, the surface current direction is reproduced fairly well by
M2D. The mean (time and space averaged) error of flood and ebb tide velocity is 0.17 m/s and 0.25 m/s, respectively, while
for the current direction, the corresponding errors are 2° and 2.5°. In general, larger discrepancies are found over sandbanks
indicating difficulties in modeling the tidal stream over complex and rapidly changing bathymetry.

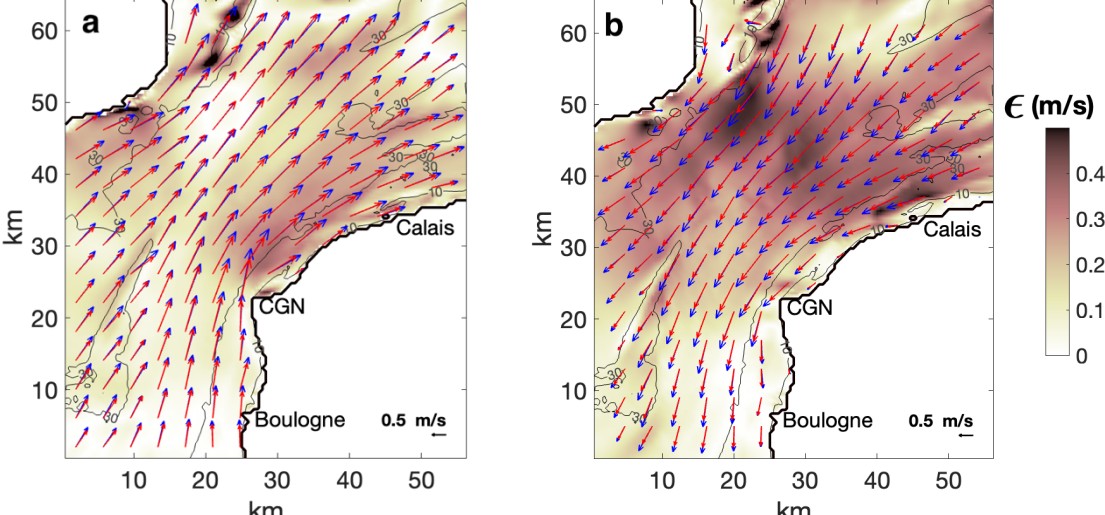

**Figure 4. Current velocities during peak flood flow (a) and peak ebb flow (b) of survey S2. Red and blue vectors show the initial and optimized model velocities respectively. The absolute difference between the initial and optimized model velocity ($|u_{OI} - u_m|$) is shown by color shading.**




The relative error of velocity, $\varepsilon$, appears significantly different in the initial model and after OI (Tab. 1). In the initial model, the error ($\varepsilon_m$) is found fairly large: 0.27 for S1 and 0.22 for S2. Blending the model with Lagrangian observations allowed to decrease the relative error by 40% for S1 and by more than 50% for S2 (Tab. 1, columns 2 and 3). Larger error obtained for S1 could be due to the location of drifter trajectories too close to the shore, during 10 hours after the release. The model performance in reconstructing the drifter trajectories is probably limited in this very shallow-water region, in the vicinity of the CGN.

It is interesting to quantify the sensibility of OI to the number of ensemble members used. While the number is limited to seven (the smallest number identified for both surveys) the results of velocity interpolation do not change much, by less than 10% (Tab. 1, column 3, values in italic). These outcomes prove that in tide dominated basins, and in the EEC in particular, the accuracy of OI is not much sensitive to the number of ensemble members used in calculating the velocity covariances. With tidal range varying between 3 and 9 meters in the EEC, the correlations of current velocities are high. This may explain why increasing the number of ensemble members results in only a slight decrease in the interpolation error.

**Table 1. Relative error $\varepsilon$ (columns 2-4) and mean (time-space averaged) separation distance (columns 5-7) between the observed and reconstructed drifter trajectories, using the initial model, optimized model, and the model after performing wind-induced velocity correction. Errors, obtained with 31 ensemble members for S1, 36 for S2, are shown in normal font, and that obtained with 7 ensemble members are given in italic.**

| | Relative error | | | Mean separation distance in km | | |
|---|---|---|---|---|---|---|
| | $\varepsilon_m$ | $\varepsilon_{OI}$ | $\varepsilon_{cor}$ | $d_m$ | $d_{OI}$ | $d_{cor}$ |
| **S1** | 0.27 | 0.16 (*0.17*) | 0.16 | 1.5 | 1.4 | 1.3 |
| **S2** | 0.22 | 0.10 (*0.11*) | 0.10 | 5.7 | 3.0 | 2.1 |

Figures 5a and 5b show the evolution of differences between the velocity provided by the initial and optimized model in drifter locations. Larger discrepancy between the observed and modeled velocities (0.2 - 0.25 m/s) is attained for $u_x$ component in the initial model during both peak flood (time 27, 39h) and peak ebb flow (time 9, 21, 33, 45h). It appears smaller (0.1 – 0.12 m/s) for $u_y$ components of the velocity vector. However, the optimization enables to reduce the mean discrepancy from 0.1 m/s for $u_x$ and 0.06 m/s for $u_y$ down to ~0.05 m/s. The discrepancy averaged over all drifters of S2 was reduced from 0.09 m/s for $u_x$ and 0.06 m/s for $u_y$ to 0.06 m/s and 0.05 m/s.





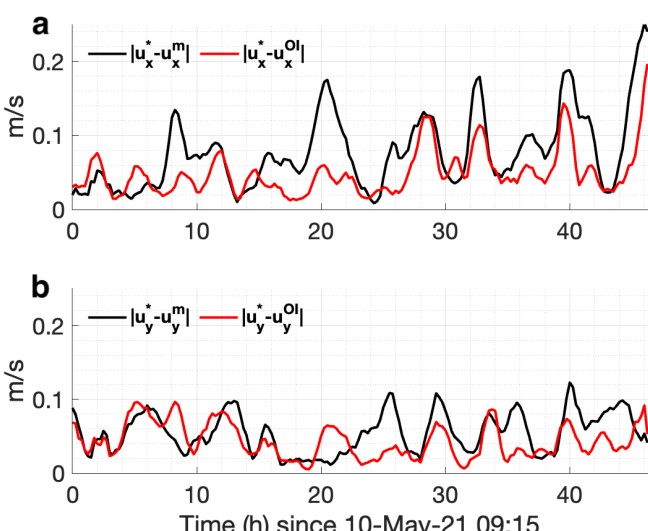

**Figure 5. Time series of absolute zonal (a) and meridional (b) velocity difference between observations and initial model (black) and observations and optimized model (red) at S2-4 drifter locations.**

Figure 6 shows the observed drifter trajectory and that provided by the initial and optimized model during S1 and S2. The corresponding separation distance, time and space averaged, is given in Table 1 (columns 5-7). During S1, the mean initial separation distance $d_m$ is 1.5 km. It decreases by 0.1 km after OI ($d_{OI} = 1.4\ km$). During S2, the model, both initial and

optimized, underestimates the northward flow component (Fig. 6b). The time evolution of the trajectory of drifter S2-4 is well reproduced but appears shifted by 4 km compared to the observed trajectory. This gives a large mean separation distance $d_m = 5.7\ km$. Blending the model with observations enables to reduce the mean separation distance by 7% for S1 and by 48% for S2 (Tab. 1, columns 4-5). However, the difference between the real and virtual drifter trajectories remains significant, especially for S2 (Fig. 6b).

Another way to evaluate the performance of OI is to perform a "cross validation" experiment. In this experiment, the optimization is done using only one drifter and the remaining drifters are advected in this optimized current field. Correcting the model velocity with observations from a single drifter reduces the model-data relative error $\varepsilon$ by 22% for S1 and by 36% for S2. These values of error reduction appear similar to those given in Table 1 and demonstrate the efficiency of the OI. This means that with only one or few drifters, it is possible to improve the model velocities in an optimal way.





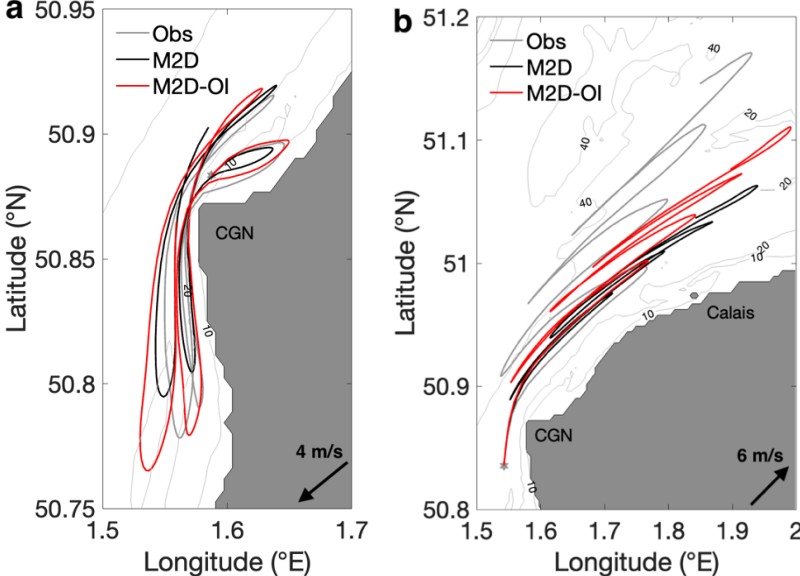

**Figure 6. Observed trajectory of drifter S1-1 (a) and S2-2 (b) (grey lines). Corresponding trajectories provided by the initial model M2D are shown in black and that resulting from OI in red. Mean wind speed and direction are shown by black arrow.**

## 4.2 Wind-induced velocity correction

The fact that larger separation distance between the observed and reconstructed trajectories was obtained during S2 (under 295 strong wind conditions) indicates that the effect of wind on surface currents is poorly reproduced in the model. To further reduce the discrepancy between the observed and modeled trajectory, the Least Squares method is used to estimate a correction to wind-induced velocity. Figure 7 shows the evolution of the separation distance between the observed and reconstructed drifter trajectories using the initial, optimized, and velocity field after correcting the wind-induced current. The correction term $c\bar{U}_{10}$ (Eq. 3) was calculated for both zonal and meridional wind components and is given as percentage of 300 the wind speed in Figure 7.

Figure 7a demonstrates that, under certain conditions, the wind-induced velocity correction is not effective. For example, at hour 14 (Fig. 7a), the separation distance attains its largest value ($d_{cor} \sim 3\ km$) for drifter S1-2. This could be due to the location of the buoy too close to the shore and to the CGN cliffs (50 m high) where the sea surface and the buoys are less exposed to the effect of northwest winds. However, the correction of the wind-induced velocity enables much better 305 trajectory reconstruction with an averaged separation distance $d_{cor} = 1.3\ km$ (Tab. 1, column 7). During S2, the separation distance $d_{cor}$ is slightly larger (~2 km) than $d_{opt}$ during the first 10 hours of drift (Fig. 7b), when the wind speed decreased from 10.5 m/s to 2 m/s. On the contrary, during the second part of the survey, when the wind increased again to ~8 m/s, the correction provides much better results, with $d_{cor}$ not exceeding 2.5 km (Fig. 7b).



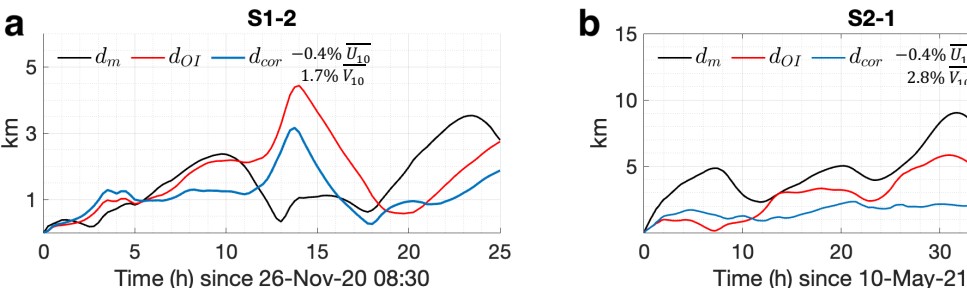

**Figure 7. Separation distance between the observed and reconstructed trajectories for drifter S1-2 (a) and drifter S2-1 (b) using the initial and optimized model, and the model after performing the wind-induced velocity correction.**

On the whole, it is remarkable that the wind-induced velocity correction enables much better trajectory reconstruction, especially during S2 (Fig. 8b), providing a total reduction of $d$ by 63%. The corresponding improvement in $d$ for S1 was limited to 13% (Tab. 1, columns 5-7) and the real and modeled trajectories appear quite similar.

Compared to the separation distance $d$, the relative error $\varepsilon$ (Tab. 1, columns 2-4) appears equal for both the optimized model and model after performing wind-induced velocity correction. Because the relative error accumulates over time, a small error does not imply the best trajectory reconstruction, either in space or time. This underlines the usefulness of separation distance for evaluating the model velocity field in terms of Lagrangian tracking.

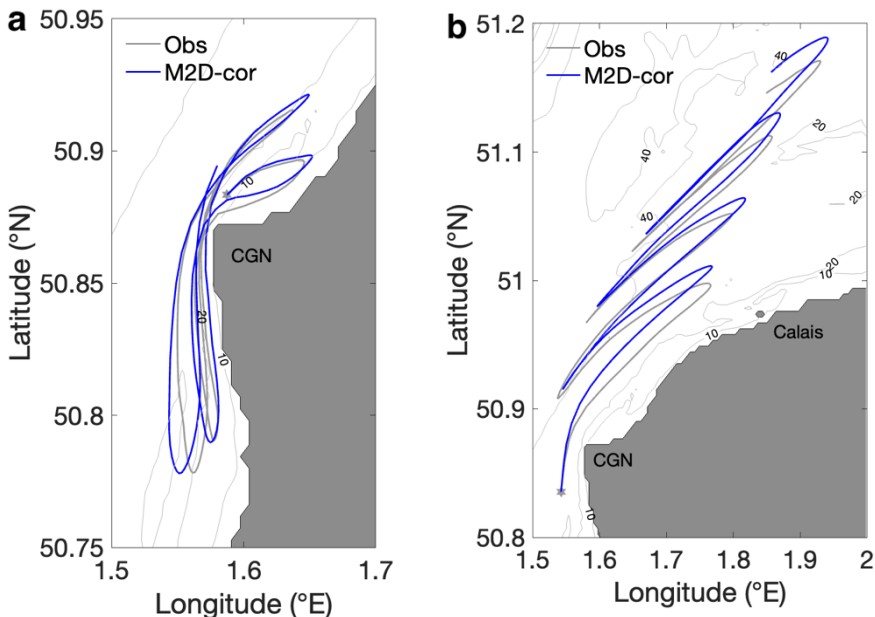

**Figure 8. Observed drifter trajectory (grey) and the trajectory reconstructed after applying wind-induced velocity correction (blue) for S1-1 (a) and S2-1 (b).**



### 4.3 Absolute dispersion

After applying OI of velocity observations and correcting the wind-induced velocity, the resulting surface currents fields are used with more confidence to assess dispersion processes, in particular by estimating the absolute dispersion. A total of 225 particles separated by 250 m were seeded within a rectangular shape area north of CGN. The center of mass of this cluster of
particles, referred to hereafter as cluster-N, was located 1.7 km offshore. The second cluster, referred to as cluster-W, was located west of CGN with its center of mass separated from the shore by 2.1 km (Fig. 9a). Each cluster formed a rectangle of size 3.3 km by 3.5 km. The particles were advected during 26-h and 46-h time period using OceanParcels software and three velocity fields provided by the initial and optimized model, and the model after performing the wind-induced velocity correction.

Absolute dispersion is used to quantify the rate of spreading. PCA allows to characterize the dominant direction of spreading and the shape of a cluster of passively drifting particles at different time intervals. Fig. 9a shows the time evolution of spreading along the ellipse axes ($A_1$ and $A_2$) during S2 at 6-h time step roughly corresponding to the time of high and low water in Boulogne. The spreading appears significantly larger in the alongshore direction. Similar results are obtained for particles in cluster-W (not shown). The effect of tidal currents on particles spreading consists in elongation of the cloud of
particles in the dominant current direction.

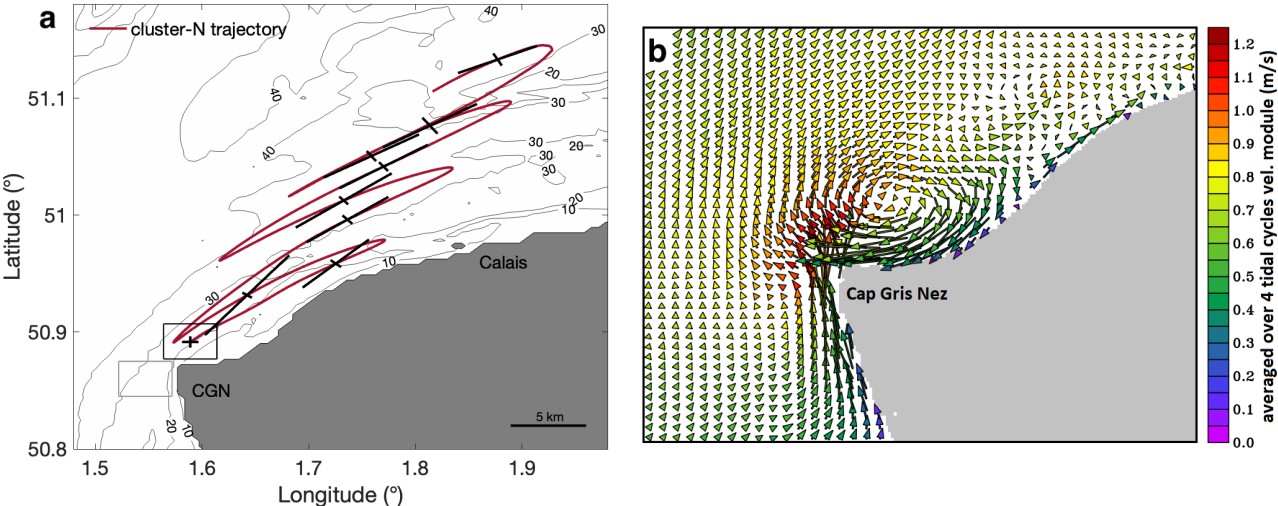

**Figure 9. (a) Evolution of the spreading along the trajectory of the center of mass of particles in cluster-N (6-h spacing) during S2. The length of semi-axis of the ellipse approximating the particle dispersion accounts for the particle spreading ($A_1, A_2$). Results were obtained with optimized surface currents after performing wind-induced velocity correction. The area of particle release is shown by black rectangle for cluster-N and grey rectangle for cluster-W. (b) Residual velocity around the CGN obtained by**
**averaging the model velocities over 4 tidal cycles.**





The time evolution of spreading during both surveys is shown in Figure 10. Similar results are obtained during S1 and S2 with four times larger spreading estimated for cluster-N than for cluster-W ($A_1 = 5.8\ km$ for cluster-N and $A_1 = 1.4\ km$ for cluster-W during S1). Particles in cluster-N experienced very large spreading shortly after the release under stronger wind

conditions observed during S2: $A_1 = 10\ km$ at time $t = 8\ h$ (Fig. 10b). The spreading is found 30% weaker ($A_1 = 7\ km$) under northeastern wind conditions (S1) with smaller wind speed (Fig. 10a). The enhanced spreading of particles in cluster-N is due to large velocity shear induced by an anticyclonic tidal eddy generated during flooding tide (Fig. 9b). Particles in this cluster are effectively driven by the eddy, whose larger nearshore velocities induce stirring of particles westward along the shore. When the tidal eddy weakens and disappears, $A_1$ slightly decreases, causing particles alignment in the main

direction of the flow. At each peak flood tide, the stronger and heterogenous tidal flow coming from the Strait of Dover towards the North Sea (mean velocity of 0.9 m/s and spatial range of variation of 1.6 m/s) causes shear dispersion and increases the spreading rate. Whereas at each peak ebb flow, the spreading along the major axis decreases. Particles seem to become more concentrated when impacted by the weaker and homogenous tidal flow (mean velocity of 0.4 m/s and spatial range of variation of 1.2 m/s). In contrast, particles in cluster-W are impacted only by the tip of the tidally induced eddy,

pushing them toward the northeast, away from the area of large nearshore velocity. Thus, particles in cluster-W experience a relatively weak spreading during both survey periods ($A_1 < 2.4\ km$).

In comparison to the model after interpolating the velocity measurements and performing the wind-induced velocity correction, the initial model tends to underestimate the spreading along the major axis $A_1$ by 20% whereas it tends to overestimate the spreading along the minor axis $A_2$ by 13% for both clusters and both surveys. It should be mentioned that

the optimized model and the model with wind-induced velocity correction provides nearly identical results (1% difference is found).

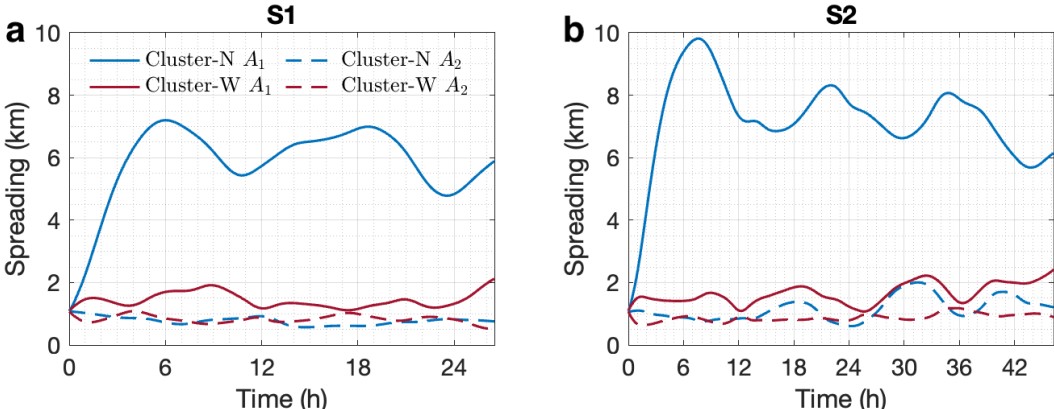

**Figure 10. Time evolution of spreading $A_1$ (solid line) and $A_2$ (dashed line) under environmental conditions observed during S1 (a) and S2 (b). Spreading of cluster-N are shown in blue and of cluster-W in dark red. Results are obtained from the model after performing wind-induced velocity correction.**



## 5 Discussion

The present study demonstrates how the Lagrangian observations can be used to improve the performance of the high-resolution model MARS. It is shown that the optimal interpolation of drifter data affects not only the model velocity fields but also the dispersion properties. Optimizing the model outputs and correcting for wind-induced velocity reduces the model-data misfit for velocity by 50% and results in a significant (10-20%) change of the dispersion rate caused by the correction of velocities.

Objective mapping methods, including OI, have been widely used in oceanographic studies. Sentchev and Yaremchuk (2015), Thiébaut et al., (2019) applied the OI to constrain a high-resolution simulation of coastal currents by MARS-2D model using towed Acoustic Doppler Current Profiler (ADCP) measurements in the English Channel. They obtained a significant decrease of the model error (50%), as the result of the velocity correction.

Kim et al. (2008) optimally interpolated the surface current velocities derived from HF radar measurements along the west coast of the United States by using a predefined (exponential shape) isotropic spatial covariance function, instead of covariance matrix derived from ensemble model simulations. The method allowed to obtain a continuous set of current vector maps by taking into account the measurement accuracy. Similar approach has been used for surface current mapping from satellite altimeter data at the global scale (e. g., Ma and Han, 2019; Wilkin et al., 2002).

The efficiency of optimal interpolation of drifter observations has been assessed in detail by Molcard et al. (2003). Using a quasi-geostrophic model within an idealized domain, interpolation scheme based on general Baysiean theory, and twin data experiments with virtual drifters, the authors quantified the performance of data assimilation. For an optimal choice of parameters (number of drifters, sampling period, and uncertainties of observations and model outputs) the relative error between the observed and modeled quantities decreased from 58% to 11%. The final model-data discrepancy obtained in our study appears to be similar (Tab. 1, column 3). This increases confidence in the results of the proposed optimization technique.

To further explore the performance of OI in application to drifter data in the tide dominated basin, sensitivity of the model correction to the number of ensemble members was assessed. The results showed that in the EEC, the performance of OI was not significantly affected by the number of ensemble members. Increasing this number from 7 to 31 provided only a 10% reduction in relative error. However, in regions with low tidal forcing (e. g., Mediterranean Sea) or with significant swell and freshwater inputs, selecting ensemble members could be more challenging. In such cases, alternative clustering methods like K-Means or SOM ( Self-Organizing Maps) could be considered (Hernández-Carrasco et al., 2018; Nguyen-Duy et al., 2021; Solabarrieta et al., 2015).

A method of correction for the wind effect, often poorly represented in numerical models, especially during the periods of strong winds, appears simple, physically robust, and efficient. A comparison of the modeled and observed drifter trajectories revealed that wind-induced velocities are largely overestimated in M2D. As a result, a significant shoreward displacement of the modeled trajectory under strong southwestern winds was obtained (Fig. 6b). The mean separation distance between the



observed and modeled trajectories attained 5 km (Tab. 1 column 5) and the maximum separation 13 km for drifter S2-3. In order to achieve better agreement, the wind-induced current velocity correction was done (Eq. 3) under the assumption of a

stationary wind over the observation period. The wind time series from ARPEGE atmospheric model and observations at meteorological stations supported this choice. In principle, the method of correction can be easily adopted for situations with evolving wind. However, in other situations, for example, when the drifters were observed close to the shore (Fig. 6a), the correction method may be less efficient. In our case, the proposed correction method allowed to reduce the separation distance between the observed and modeled trajectories by 63% for S2, under strong winds, and by 13% for S1, under weak

winds (Tab. 1, column 7).

This highlights the importance of an accurate representation of the wind effect in high-resolution coastal circulation models. For example, the effect of Stokes drift on passive tracers, drifting in the surface layer, should be accounted for. In fact, MARS is Eulerian hydrodynamic model, not coupled with a wave model in the considered configuration. For this reason, wave-current interactions are neglected in the model. Moreover, the wave-induced current velocity (Stokes drift velocity),

estimated as 1% of the wind speed (Ardhuin et al., 2018, 2012), can modify considerably the transport pathways of passively advected particles. Dobler et al. (2019), Van den Bremer and Breivik (2018), Curcic et al. (2016) also highlighted the impact of Stokes drift on the behavior of passive tracers, micro-plastics, or oil spills, especially under strong winds.

One of the practical applications of oceanographic studies is the assessment of turbulent dispersion of materials in the marine coastal environment. It attracts a growing interest because our seas and oceans are being degraded by human activities that

harm marine life, undermine coastal communities, and inject harmful substances into the ocean (Landrigan et al., 2020). Marine turbulence is considered the main factor controlling the spreading of materials in seawater (van Sebille et al., 2020). The present study aims to evaluate the turbulent dispersion and demonstrate how the dispersion estimates can be improved in one of the busiest maritime straits. Optimal interpolation of drifter data was used to optimize the sea current velocities. It was found that the resulting change in the velocity field may lead to adjustment of the velocity gradients which, in turn, increase

the rate of dispersion. Consequently, the absolute dispersion in the model was found to be significantly larger after interpolation of the drifter data, which is not surprising given the results reported in other studies. Modeled velocities are generally lower and less variable than observed velocities (Kjellsson and Döös, 2012), especially under strong wind conditions (Curcic et al., 2016).

In addition, other studies highlighted that in tide-dominated regions, with large spatial variation of velocity, the coastal flow

is characterized by strong shear dispersion (Van Dam et al., 1999; Zimmerman, 1986). In particular, an enhancement of the dispersion rate was found in the vicinity of headlands, or under a significant bathymetric change (Geyer and Signell, 1992). Numerical studies in the English Channel have shown that passive particles released offshore experience lower dispersion compared to the particles released close to the shore where the bathymetry variation is large. Sentchev and Korotenko (2005) documented that under the joint effect of freshwater input and tides, a cluster of particles released in the nearshore coastal

flow experienced large stretching along the shore. These results are in good agreement with the behavior of particles in cluster-N, affected by the near-shore coastal flow and tide-generated transient eddy.





## 6 Conclusions

In this study, we tested a computationally efficient method of combining numerical modeling with surface drifter
observations to obtain a more reliable estimate of turbulent dispersion in the narrowest and most energetic part of the EEC -
the Dover Strait.

Using optimal interpolation to combine the high-resolution MARS model outputs with two and four drifter trajectories
allowed reconstruction of the surface velocity evolution with a 50% reduction in the error between observed and modeled
velocities. Additional correction of the wind-induced velocity component enabled to further reduce the separation distance
between observed and modeled trajectories (63% reduction of separation distance under strong winds). Particle spreading,
estimated from more realistic velocities, obtained after the OI and wind-induced current corrections, was found 20% higher
north of the CGN and 13% lower south of the CGN, compared to the initial model run. Spatial variability in dispersion was
identified. It is assumed to be related to small-scale features of the local circulation generated by tidal flow interaction with
the headland (CGN) and irregular topography.
The proposed methodology can be used to improve existing decision-making support tool or design new tools for monitoring
the transport and dispersion of materials in coastal ocean environment.

## Code availability

The Optimal Interpolation code and the scripts to reproduce the figures of the manuscript are available upon request to the
corresponding author.

## Data availability

MARC simulations from MARS numerical ocean model are available on the project website at https://marc.ifremer.fr.
Meteorological data from MeteoFrance (observations and model) are now available at:
https://publitheque.meteo.fr/okapi/accueil/okapiWebPubli/index.jsp.

## Author contributions

AS designed the drifters experiments and carried them out. SB performed the data analysis and prepared the manuscript with
contributions from all co-authors.

## Competing interests

The authors declare no conflict of interest.



**Acknowledgments**

The Ph.D thesis of Sloane Bertin has been cofounded by the Région Hauts de France and the University of Littoral Côte

d'Opale. This research has been funded by the French National program LEFE (Les Enveloppes Fluides de l'Environnement). The authors would like to warmly thank Eric Lecuyer for creating the home-made drifters used in the study and his help during sea trials. We thank Maxime Touchais and all people who helped with the drifter deployment and recovery. Fruitful collaboration with Max Yaremchuk is acknowledged.

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
