# Peer review of "Fusion of Lagrangian drifter data and numerical model outputs for improved assessment of turbulent dispersion"

_EGUsphere, 2024_

## Referee Comment (RC1)

This research used Lagrangian drifter derived surface velocity to improve the numerical model simulated ocean velocity field, which is important for turbulence and transport studies at sub-mesoscale. The results show that there is a significant improvement of the model capability after the optimal interpolation using drifter-derived velocity and wind-induced velocity correction. However, the manuscript still has some flaws. For example, the long simulation settings are not detailed. The modeled results at all drifters of S1 and S2 are not presented and described in the manuscript. Therefore, this study may be accepted for publication in EGUsphere after minor revisions.

**Major comments:**

1. Line 125 in page 5, authors just present the tidal conditions for model's boundary. What are the other boundary conditions for simulation? Such as gradient, Clamped, Flather or other boundary conditions?

2. Two one-year long model runs are implemented in this research. Can authors provide more detailed modeling settings to make model stable in such long simulation.

3. In Figure 5, authors just present the model absolute errors at S2-4. May authors present the absolute errors using "Box-Whisker" plot over all drifters of S2?

4. Line 280-281, authors describe that the drifter S2-4 is well reproduced. Can authors plot all drifters of S1and S2 in Figure 6 to make reader directly understand the simulation results. There is similar problem in Figure 7. In Figure 7, authors use drifters S1-2 and S2-1, why is it different from Figure 6? Can authors also present all drifters of S1 and S2 in Figure 7? Or using "Box-Whisker" plot to present separation distance in Figure 7? Furthermore, how about the wind-corrected trajectories of other drifters in S1 and S2? Can present other drifters in Figure 8? If other OI- or corrected trajectories are similar to S1-1 and S2-1, please describe the related statement of other trajectories in S1 and S2.

**Minor comments:**

1. Line 121 in page 5, the temporal resolution is 15 min, which is model output temporal resolution or model simulation temporal resolution?

2. Line 122-124 in page 5, authors provided the accuracy of MARC simulation results. Please provide a reference for this statement.

3. Line 158 in page 7, it should be the background model "velocity" $u_m$?

4. Line 161 in page 7, please describe the meaning of primes in $u_m(x', t')$.

5. In Figure 3, please describe the meanings of blue dash and solid lines.

6. Line 253 in page 11, should add a comma after "seven (...)".

7. Line 269-274 in page 11, authors should point out the discrepancy results in Figure 5 are from drifter S2-4.

8. In Figure 6, authors can change the lines' color of observed trajectories to easily distinguish the observed, M2D, and M2D-OI results. Such as blue lines for observed trajectory.

9. Line 306 in page 13, $d_{opt}$ should be revised to $d_{OI}$?

10. In caption of Figure 9, please describe the meaning of red trajectory. Meanwhile, authors should point out the semi-axis of the ellipse is represented by the black lines.

---

## Author Comment (AC4)

Dear Reviewer,

We greatly appreciate your constructive comments. Below we provide detailed point-by-point responses as well as changes performed in the new version of the manuscript regarding the comments. Modifications in the revised manuscript are shown in purple.

1. **Discussion of 3D Processes**: The study is based on a 2D model, which ignores all baroclinic processes. It is necessary to include a brief discussion in section 2.1 about 3D processes, with references, and clearly state that 3D processes can be ignored compared to tidal currents.

*Thank you for your pertinent comment. Following the Reviewer's recommendation a text containing a discussion of the choice of 2D model for the present study has been added to section 2.3, lines 115-119. For your convenience, the text is also reproduced bellow.*

The water dynamics in the eastern English Channel is largely dominated by tides. The baroclinic effects on the vertical are negligible due to the enhanced mixing affecting the entire water column (e.g., Breton and Salomon, 1995). Moreover, the study area is located fairly far away from the major source of buoyancy – the Seine River, whose discharge was low during the measurement period. The use of a 2D model was therefore justified. The variation of salinity in the horizontal plan is taken into account in the 2D model.

2. **Stokes Drift**: You mention that the Stokes drift impacts the drift and is ignored. You need to provide estimated values of surface Stokes drift based on wind speed or available modeled results (e.g., WWIII results from Ardhuin Fabrice's group).

*The estimation of surface Stokes drift based on wind speed is given in the discussion (Section 5), L472-474 of the revised manuscript.*

3. **Drifter Parameters**: You listed some parameters of the drifter used in this study. However, it is unclear whether the drifter measured surface trajectories or averaged depth trajectories. Please clarify this.

*Thank you for your comment. The presentation of drifters has been clarified in Section 2.2, L105 of the revised manuscript.*

4. **Relevance of Paragraph L30-L40**: This paragraph does not seem closely related to the study. Your study is mostly based on a 2D model focusing on tidal dynamics in coastal areas. The paragraph discusses mesoscale and submesoscale processes in the ocean, which are mainly associated with 3D baroclinic processes. Consider revising or removing this paragraph for better alignment with the study's focus.

*Following the Reviewer's recommendation, the corresponding paragraph in Introduction has been removed.*

---

## Author Response (AR1)

**Response to Reviewers**

Manuscript ID: egusphere-2024-176
Title: Fusion of Lagrangian drifter data and numerical model outputs for improved assessment of turbulent dispersion
Journal: Ocean Science, Special Issue for the 54th International Liège Colloquium on Machine Learning and Data Analysis in Oceanography
Authors: S. Bertin, A. Sentchev, E. Alekseenko

Dear Editor,

Please find enclosed the revised version of the manuscript which we modified according to the Reviewer's suggestions.

We greatly appreciate the constructive comments from all the Reviewers. Below we provide detailed point-by-point responses as well as changes performed in the new version of the manuscript regarding the comments.

Reviewers' comments are shown in regular and our responses in italic font. Line numbers and figures indicated in our response refer to the new version of the manuscript unless otherwise noted. In the annotated version of the manuscript, the main changes performed are given in red for Reviewer #1, blue for Reviewer #2, green for Reviewer #3 and purple for Reviewer #4.

We warmly thank all the Reviewers for the constructive comments, and we feel that the manuscript has significantly improved. We hope that our response will satisfy them.

With kind regards,

Sloane Bertin, on behalf of all authors.

**REVIEWER #1**

**Major comments**

1. Line 125 in page 5, authors just present the tidal conditions for model's boundary. What are the other boundary conditions for simulation? Such as gradient, Clamped, Flather or other boundary conditions?

*The numerical model utilizes nested configurations with progressive resolutions: (i) 2 km covering the Northeast Atlantic (level 0), (ii) 700 m at the regional scale, encompassing the English Channel (level 1), and (iii) 250 m for the Eastern English Channel (level 2). This nesting technique enables the accurate capture of interactions between large-scale and small-scale processes. This enables the transfer of all resolved fields from lower resolution levels to the open boundaries of higher resolution levels.*

*The model accounts for kinematic free-surface and bottom boundary conditions, contingent upon friction terms (Lazure and Dumas, 2008). The turbulence closure employed in the model follows the approach described in Gaspar et al. (1990).*

*All these details will be added to the part 2.3 'Current velocity from numerical model'.*

2. Two one-year long model runs are implemented in this research. Can authors provide more detailed modeling settings to make model stable in such long simulation.

*Comprehensive information regarding model equations, the coupling of barotropic and baroclinic modes, model discretization, solving methods, computational stability according to CFL criterion (table 1, Lazure and Dumas, 2008), and costs are meticulously outlined in Lazure and Dumas (2008). To maintain CFL stability, the modeling timestep was set to 30 seconds for the level 2 model.*

*All these details will be added to the part 2.3 'Current velocity from numerical model'.*

3. In Figure 5, authors just present the model absolute errors at S2-4. May authors present the absolute errors using "Box-Whisker" plot over all drifters of S2?

*Thank you for this advice. I modified Figure 5 and used box-whiskers to present the model absolute error for all the drifters during S1 (Fig. 5a) and S2 (Fig. 5b). Consequently, I modified the descriptive paragraph L301-308 and legend.*

4. Line 280-281, authors describe that the drifter S2-4 is well reproduced. Can authors plot all drifters of S1and S2 in Figure 6 to make reader directly understand the simulation results. There is similar problem in Figure 7. In Figure 7, authors use drifters S1-2 and S2-1, why is it different from Figure 6? Can authors also present all drifters of S1 and S2 in Figure 7? Or using "Box-Whisker" plot to present separation distance in Figure 7? Furthermore, how about the wind-corrected trajectories of other drifters in S1 and S2? Can present other drifters in Figure 8? If other OI- or corrected trajectories are similar to S1-1 and S2-1, please describe the related statement of other trajectories in S1 and S2.

*I modified Figures 6 and 8 by presenting all the drifters' trajectories (2 for S1 and 4 for S2) in order to make the reader directly understand the simulation results. I also modified Figure 7 by using box-whiskers to present the separation distance results for all the drifters simultaneously. Consequently, the paragraphs and legends concerning these three figures have been modified.*

**REVIEWER #2**

**Major comments:**

1. Line 106 in page 5, the author mentions that all drifters were equipped with an anchor to allow them to drift with surface currents. However, it is not specified whether there were

sensors to monitor the presence of the anchor or if there was an assessment of the anchor's stability in the marine environment.

*Thank you for your comment. There were no sensors to monitor the presence of anchor on the buoys. All the buoys were recovered at the end of each campaign, which was limited to one or two days due to the high speed of currents in the eastern English Channel. No one drifter lost its anchor. We deployed our drifters for longer periods in other geographical areas. The anchor was present again after recovery. This proves that the equipment we used is of good quality.*

*Concerning the stability of the drogue at sea, we used professional equipment, anchors originally designed for sailing boats. Prior to the campaign, we monitored drifters with anchors at sea, and never noticed any problem with the anchors. It is worth noting that the Nomad drifters, manufactured by SouthTek (https://www.southteksl.com), are equipped with exactly the same anchors.*

2. In the process of optimizing model evaluation, this study extensively utilized fused data sources to assess fusion outcomes. However, such an evaluation process may not objectively reflect the effectiveness of the fusion method and the characteristics of the real ocean current field. Given the scarcity of high-resolution observational data in the study area, buoy data can be partitioned into training and validation sets. The "cross validation" method mentioned at line 285 is an effective approach for dataset partitioning, which could be considered as a core method to extend across various stages of model evaluation, illustrated through figures and charts.

*We thank the Reviewer for this comment which is in line with the comment 2 of the Reviewer #3. We agree that using fused data to assess the results of fusion may not always objectively reflect the effectiveness of the fusion method. However, "validation without exclusion" (see response below to major comment 3) enables to assess the quality of our results and the effectiveness of the fusion method objectively.*

*In addition, and again we agree with the Reviewer on this point, the "cross-validation" method is probably the best way to evaluate the performance of the data fusion technique. However, it requires larger amount of high-resolution observational data (more drifters, or remotely sensed data, for example) that we did not have. A paragraph has been added to the discussion in the revised version of the manuscript (lines 444-449) to address this issue.*

3. Line 285 in page 12, the author mentioned the "cross validation experiment," where one drifter was used for model optimization and the others for validation. However, the cross-validation method imposes high requirements on the randomness and independence between the training and validation sets, often employing random sampling. Can simply selecting one drifter as the training set meet these requirements? For example, considering that drifters released during the same period exhibit highly similar and repetitive trajectories due to minimal differences in release times and geographic distances, would the cross-validation method remain effective in such scenarios?

*We agree with the Reviewer on this point. We did not use a random set for training; therefore, the term "cross-validation" is not appropriate. To meet the Reviewer recommendation, we applied another technique of validation, which is "leave-one-out validation". It provides a much less biased measure of error compared to the previously used method of validation, because we repeatedly fit the model to a dataset that contains n-1 drifter trajectories. More specifically, the method involves using one drifter trajectory as a control data set and other trajectories for optimization. The control trajectory is repeatedly replaced during the validation exercise. At the end, the mean relative error of optimization was reduced by 22% for S1 and by 36% for S2. A new text describing the validation technique and the results has been added on page 14 (lines 325-330) of the revised manuscript.*

*Even if the drifters are released with minimal differences in time and geographic distances, their trajectories are different, in particular, in the Cap Griz Nez region, characterized by a complex current structure and large velocity (see Fig. 9b). This variability ensures that the leave-one-out validation method remains effective. The observational data set provide a solid basis for both optimization and validation, demonstrating that the model improvements are not simply coincidental but rather the result of effective capturing of the underlying physical processes.*

**Minor comments:**

1. Line 103 in page 5, the construction of laboratory-made drifters was described. It would be beneficial to also introduce the construction of Nomad drifters and provide a comparison between these two types of drifters.

*Thank you for your pertinent comment. The description of the coastal Nomad drifters manufactured by SouthTek was added line 106-107 of the revised manuscript. For this study, we assumed that differences between the two types of drifters were negligible, which was added line 111-112 of the revised manuscript.*

2. Line 106 in page 5, the author mentioned that all drifters were equipped with an anchor of 0.5 m long positioned in the water layer between 0.8 and 1.3 m depth. Would it be feasible to calculate the overall center of buoyancy depth, including the anchor?

*Thank you for this remark, the overall center of buoyancy depth has been estimated to 1 m and this has been modified in the revised manuscript.*

3. Line 108 in page 5, the author mentions that observed surface current velocities were estimated from the drifter trajectories. Please describe the specific method used.

*Thank you for your comment, this has been clarified in the revised manuscript lines 108-109.*

4. In Figure 1, would it be better to align the display area of Figure 1b with the measurement area outlined in Figure 1a?

*Thank you for your suggestion. We modified Figure 1b to show that it corresponds to the red rectangle in Figure 1a.*

5. Issues with image consistency. For example, the image sizes and font sizes of axis labels in Figures 1a and 1b are inconsistent. Additionally, the positioning of subplot identifiers in Figures 1 and 9 is inconsistent.

*This has been addressed in the revised manuscript, thank you for your pertinent remark.*

**REVIEWER #3**

1. Authors state that "The wind significantly affects the local circulation" in the region. However, results indicate that the model reconstruction is not sensitive to the number of ensembles used (with ensembles generated from perturbing wind forcings). Does this indicate that the assimilation scheme is not sensitive to the number of ensembles used or that perturbing winds do not significantly impact model trajectory? The domain is described as a "tide dominated basins" in line 254 but later results suggest that modifying the wind-forcing scheme has a significant impact on model outputs.

*We agree with the reviewer on this point. In fact, circulation in the Eastern Channel is dominated by a combination of wind and tides, which is supported by the results of our study. The two major forcing terms are taken into account when selecting ensemble members. This is why; after correcting the wind-induced velocity, the relative error does not change (Tab.1, columns 3,4), which also proves that the model trajectory is effectively modified by OI. The only one quantity affected by this correction is the separation distance during S2 (columns 6,7). According to the Reviewer's comment, the text in lines 285-287 has been modified.*

2. There is a concern that the data being assimilated into the model is also being used to evaluate the performance of the model. Does this adversely impact the evaluation and bias towards assimilation schemes that weigh more heavily towards observation data? A cross-validation scheme is implemented where only one drifter is assimilated but it isn't obvious that these are independent. Are there other independent datasets such as in-situ sensors that could be used to confirm the robustness of the findings?

*Thank you for your remark. This point has also been raised by Reviewer #2 and has been addressed. In accordance with the Reviewer's request, we changed the method of validation from "cross-validation" to "leave-one-out validation". Leave-one-out validation provides a much less biased measure of relative error compared to that used in cross-validation, because we repeatedly fit a model to a dataset that contains n-1 drifter trajectories. A new text*

*describing the validation technique and the results has been added on page 14 (lines 325-330) of the revised manuscript.*

*Regarding the second part of the Reviewer's comment, a text has been added in section 5 (lines 444-450) Unfortunately, there were no high-resolution data in the study area during the drifters' deployment.*

3. Does the wind correction scheme in (3) require information on drifter & model data for the entirety of the period? i.e. are corrections being made at time t using information available at time t+1?

*The wind correction scheme uses data (drifter velocities and the model counterparts) for all time steps, i.e. for the entire period of measurements. Therefore, the correction coefficient c is obtained for the whole study area and the survey period. This is explained in lines 242-244. Nevertheless, it is possible to obtain a time-varying correction coefficient. This was not our strategy given the short duration of the observation period.*

**REVIEWER #4**

1. **Discussion of 3D Processes**: The study is based on a 2D model, which ignores all baroclinic processes. It is necessary to include a brief discussion in section 2.1 about 3D processes, with references, and clearly state that 3D processes can be ignored compared to tidal currents.

*Thank you for your pertinent comment. Following the Reviewer's recommendation a text containing a discussion of the choice of 2D model for the present study has been added to section 2.3, lines 115-119.*

2. **Stokes Drift**: You mention that the Stokes drift impacts the drift and is ignored. You need to provide estimated values of surface Stokes drift based on wind speed or available modeled results (e.g., WWIII results from Ardhuin Fabrice's group).

*The estimation of surface Stokes drift based on wind speed is given in the discussion (Section 5), L472-474 of the revised manuscript.*

3. **Drifter Parameters**: You listed some parameters of the drifter used in this study. However, it is unclear whether the drifter measured surface trajectories or averaged depth trajectories. Please clarify this.

*Thank you for your comment. The presentation of drifters has been clarified in Section 2.2, L105 of the revised manuscript.*

4. **Relevance of Paragraph L30-L40**: This paragraph does not seem closely related to the study. Your study is mostly based on a 2D model focusing on tidal dynamics in coastal

areas. The paragraph discusses mesoscale and submesoscale processes in the ocean, which are mainly associated with 3D baroclinic processes. Consider revising or removing this paragraph for better alignment with the study's focus.

*Following the Reviewer's recommendation, the corresponding paragraph in Introduction has been removed.*